# miR-877-5p as a Potential Link between Triple-Negative Breast Cancer Development and Metabolic Syndrome

**DOI:** 10.3390/ijms242316758

**Published:** 2023-11-25

**Authors:** Juana Moro, Agustina Grinpelc, Paula Lucía Farré, Rocío Belén Duca, Ezequiel Lacunza, Karen Daniela Graña, Georgina Daniela Scalise, Guillermo Nicolás Dalton, Cintia Massillo, Flavia Piccioni, Federico Dimase, Emilio Batagelj, Adriana De Siervi, Paola De Luca

**Affiliations:** 1Laboratorio de Oncología Molecular y Nuevos Blancos Terapéuticos, Instituto de Biología y Medicina Experimental (IBYME-CONICET), Buenos Aires 1428, Argentina; 2Centro de Investigaciones Inmunológicas Básicas y Aplicadas (CINIBA), Facultad de Ciencias Médicas, Universidad Nacional de La Plata, Buenos Aires 1900, Argentina; 3Laboratorio de Inmunobiología del Cáncer, Instituto de Investigaciones en Medicina Traslacional (IIMT), Universidad Austral, CONICET, Buenos Aires 1629, Argentina; 4Hospital Militar Central, CABA, Buenos Aires 1426, Argentina

**Keywords:** breast cancer, metabolic syndrome, miRNAs

## Abstract

Metabolic syndrome (MS) is a risk factor for breast cancer (BC) that increases its aggressiveness and metastasis. The prevalence of MS is higher in triple-negative breast cancer (TNBC), which is the molecular subtype with the worst prognosis. The molecular mechanisms underlying this association have not been fully elucidated. MiRNAs are small, non-coding RNAs that regulate gene expression. Aberrant expression of miRNAs in both tissues and fluids are linked to several pathologies. The aim of this work was to identify circulating miRNAs in patients with alterations associated with MS (AAMS) that also impact on BC. Using microarray technology, we detected 23 miRNAs altered in the plasma of women with AAMS that modulate processes linked to cancer. We found that let-7b-5p and miR-28-3p were decreased in plasma from patients with AAMS and also in BC tumors, while miR-877-5p was increased. Interestingly, miR-877-5p expression was associated with lower patient survival, and its expression was higher in PAM50 basal-like BC tumors compared to the other molecular subtypes. Analyses from public databases revealed that miR-877-5p was also increased in plasma from BC patients compared to plasma from healthy donors. We identified IGF2 and TIMP3 as validated target genes of miR-877-5p whose expression was decreased in BC tissue and moreover, was negatively correlated with the levels of this miRNA in the tumors. Finally, a miRNA inhibitor against miR-877-5p diminished viability and tumor growth of the TNBC model 4T1. These results reveal that miR-877-5p inhibition could be a therapeutic option for the treatment of TNBC. Further studies are needed to investigate the role of this miRNA in TNBC progression.

## 1. Introduction

Breast cancer (BC) remains one of the most important public health issues in the world [1]. Triple-negative breast cancer (TNBC) is the molecular subtype with the worst prognosis and the fewest therapeutic options [2,3].

Non-hereditary factors have a major impact on the risk and progression of BC [4,5]. Metabolic syndrome (MS) is a group of physiopathological disorders characterized by the presence of three or more of the following conditions: abdominal obesity (waist circumference ≥ 88.9 cm in women) [6], hyperglycemia (≥110 mg/dL), blood triglycerides ≥150 mg/dL (women), HDL-C cholesterol <50 mg/dL (women) and elevated blood pressure (≥130/85 mmHg) [7]. Previous studies suggest that several components of MS such as abdominal obesity, hyperglycemia, hypertension and dyslipidemia are associated with BC development [8,9]. In addition, MS is a risk factor for BC that increases its aggressiveness and metastasis [10,11,12,13,14]. Interestingly, a retrospective study found that MS was more prevalent in TNBC compared with the other PAM50 molecular subtypes [15]. Several mechanisms have been proposed to explain the association between MS and BC; however, the molecular mechanisms involved have not been fully elucidated [16]. As the prevalence of MS increases worldwide, it is critical to understand the mechanisms responsible for the effects of MS on cancer development and progression.

MicroRNAs (miRNAs) are non-coding RNAs 21–25 nucleotides long that are regulators of gene expression and modulate numerous biological processes important for maintaining homeostasis. In addition to tissue expression, miRNAs can be detected in various body fluids such as blood and urine, making them interesting biomarkers for the diagnosis and prognosis of diseases [17]. Moreover, circulating miRNAs constitute important components in cell communication at both paracrine and endocrine levels [18,19]. In fact, miRNAs’ aberrant expression, in both tissues and body fluids, has been associated with numerous pathologies, such as cancer and MS [20,21].

The aim of this work was to identify circulating miRNAs in patients with clinical features linked to MS that also impact on BC development and constitute possible therapeutic targets for these patients. Using microarray technology, we detected 23 miRNAs altered in the plasma of women with alterations associated with MS (AAMS) that modulate processes linked to cancer. We found that let-7b-5p and miR-28-3p were decreased in plasma from patients with AAMS and also in BC tumors, while miR-877-5p was increased in plasma from patients with AAMS and in BC tissue. Interestingly, miR-877-5p expression was associated with lower patient survival, and its expression was higher in basal-like BC tumors compared to the other molecular subtypes, so we focused our research on this miRNA. Thus, we identified IGF2 and TIMP3 as validated target genes of miR-877-5p whose expression is decreased in BC tissue and moreover, is negatively correlated with the levels of this miRNA in the tumors. Finally, we demonstrated that miR-877-5p inhibition caused a decrease in viability of 4T1 TNBC cells. Even more, a single injection *in vivo* with a miR-877-5p inhibitor diminished tumor growth of 4T1-derived tumors.

## 2. Results

### 2.1. Circulating miRNAs Were Altered in Plasma of Women with AAMS

To identify the circulating miRNAs’ expression profile induced by AAMS, patients from Hospital Militar Central (CABA) were recruited and divided into two groups: control and women with AAMS. The latter was defined when participants presented two or more of these conditions: BMI ≥ 25.00 kg/m^2^, waist circumference ≥ 82 cm or high blood pressure (systolic ≥ 120, diastolic ≥ 80). MiRNAs were isolated from plasma, and four samples (two control and two AAMS) were generated by pooling miRNAs from nine donors and hybridized to GeneChip^®^ miRNA 4.0 Array (Affymetrix). Following data normalization, we established a threshold for identifying significant genes, requiring a false discovery rate value (FDR) < 0.05%, Log Fold Change (LogFC) > 1.5, and a *p*-value < 0.01. Thus, we identified 23 miRNAs differentially expressed in the plasma of women with AAMS compared to the control group (Figure 1A, Table 1 and Appendix A).

To elucidate the biological role of these miRNAs, we identified the KEGG pathways to which the validated target genes of these miRNAs belong using DIANA miRPath. We found that these miRNAs participate in several processes linked to cancer such as cell cycle, proteoglycans, transcriptional misregulation and pathways in cancer (Figure 1B). In particular, we found that AAMS-downregulated miR-23a-3p, -19b-3p, -181a-5p, -122-5p, -425-5p, -28-3p, -146a-5p, let-7b-5p and AAMS-upregulated miR-101-5p and -877-5p showed the strongest association, so we focused on these miRNAs for further studies (Figure 1C).

### 2.2. miR-28-3p, -101-5p and let-7b-5p Were Diminished While miR-181a-5p, -425-5p and -877-5p Were Increased in Primary Tumors of BC Patients

Our hypothesis was that circulating miRNAs of women with AAMS could impact on BC development and progression. We assessed the role of those miRNAs in BC using public databases. Thus, we compared their expression levels in a primary tumor (PT) versus ANT from BC patients in TCGA BRCA. We found that miR-28-3p, -101-5p and let-7b-5p were diminished in PT compared to ANT while miR-181a-5p, -425-5p and -877-5p were increased (Figure 2A). Interestingly, miR-28-3p and let-7b-5p expression levels, which are diminished in AAMS patients, were also diminished in PT compared to ANT, and miR-877-5p expression levels, which are augmented in AAMS patients, were increased in PT compared to ANT.

Next, we studied the expression of these miRNAs according to PAM50 molecular subtypes in tumors from the TCGA-BRCA. These miRNAs were differentially expressed in tumors of BC patients according to PAM50 subtype (Figure 2B). Additionally, miR-877-5p expression was increased in the basal-like subtype, which is the most aggressive BC subtype, and let-7b-5p levels were diminished in basal-like tumors compared to luminal B (Figure 2B).

### 2.3. The miR-877-5p Expression Was Correlated with Diminished Overall Survival of BC Patients

We then examined the role of miR-28-3p, let-7b-5p and miR-877-5p expression in the overall survival, disease-specific survival, progression-free interval and relapse-free interval of patients with BC from TCGA-BRCA—IlluminaHi Seq using UCSC Xena tool (https://xena.ucsc.edu/) (Figure 3A–D). We found that miR-877-5p expression correlated with poor overall survival in BC patients (Figure 3A). Hence, we focused on miR-877-5p for further studies.

Also, considering TNBC is the molecular subtype with the worst prognosis and the fewest therapeutic options [2,3], we explored miR-877-5p expression levels in PAM50 basal-like tumors from the TCGA BRCA dataset, which represents around 84% of TNBC [22,23]. We found that miR-877-5p expression was increased in PAM50 basal-like breast tumors compared to ANT (Figure 4), which was reasonable considering the expression of this miRNA was increased in PAM50 basal-like tumors compared to the other molecular subtypes (Figure 2B).

### 2.4. miR-877-5p Was Increased in Plasma of AAMS Patients and BC Patients

To validate our microarray findings, we performed miRNA RT-qPCR analysis of miR-877-5p expression using the same cohort of nine women with AAMS and nine women without AAMS, as shown in Figure 1. Our miRNA RT-qPCR experiments demonstrated a significant increase in miR-877-5p expression in plasma from patients with AAMS compared to the control group (Figure 5A), providing additional support for the microarray results.

Moreover, we assessed miR-877-5p expression in blood samples from BC patients using the GSE73002 dataset. Our analysis revealed a significant increase in miR-877-5p expression in serum from patients with BC compared to healthy donors (balanced N = 1280) (Figure 5B). These results provide additional support for our previous findings and suggest that miR-877-5p may be a useful biomarker for BC as well as AAMS.

### 2.5. miR-877-5p Expression Was Inversely Correlated with IGF2, ERBB2, TIMP3 and COL6A2 Levels in Breast Tumors

To underscore the molecular mechanism by which miR-877-5p could impact on BC development and/or progression, we identified the validated target genes of miR-877-5p that are involved in pathways related to cancer using Diana miRPath (Table 2).

Next, we determined if the expression of miR-877-5p negatively correlates with its targets. The findings revealed a significant negative correlation between the expression levels of ERBB2, TIMP3, COL6A2 and IGF2 (Figure 6A). The correlation coefficient (r_s_) indicated a weak to moderate association between these genes. This classification of weak to moderate correlation coefficients (r_s_) was based on the criteria established by Guilford and Rowntree (weak: r_s_ between −0.2 and −0.4; moderate: r_s_ between −0.4 and −0.7) [24,25,26]. Additionally, using the STRING database, we identified interactions among these genes (Figure 6B). This is reasonable considering all genes are involved in related pathways such as Proteoglycans in cancer (ERBB2, TIMP3 and IGF2) and ECM–receptor interaction (COL6A2). In particular, the STRING network revealed evidence of associations between COL6A2 and TIMP3 in text mining and co-expression studies, while text mining studies associate ERBB2 and IGF2 and suggest they also share protein homology (Figure 6B). Moreover, we found that the expression of TIMP3 and IGF2 were diminished in the PT of BC patients from the BRCA-TCGA compared to ANT, while ERBB2 expression was increased and COL6A2 was not differentially expressed in these tissues (Figure 6C).

These results suggest that miR-877-5p/TIMP3/IGF2 expression in BC tissue and/or plasma could be a relevant molecular axis for BC development and progression, with high relevance in patients with AAMS.

### 2.6. miR-877-5p Inhibition Decreased Viability and Tumor Growth from the TNBC Model 4T1

We examined if miR-877-5p modulated the viability of the TNBC 4T1 cell line. The transient transfection of 4T1 cells with a miR-877-5p inhibitor diminished the expression levels of this miRNA after 24 and 48 h (Figure 7A,B). We found that the inhibition of miR-877-5p caused a decrease in viability (Figure 7C).

Finally, to analyze the effect of miR-877-5p on tumor growth of TNBC, balb/c mice were inoculated s.c. with 4T1 cells. On day 13 post-inoculation, when the tumors were around 80 mm^3^, the animals were inoculated i.p. with a single doses of PEI nanoparticles containing 0.73 nmol of miR-877-5p inhibitor or its negative control NC5 (NC). Interestingly, after 4 days, a significant decrease in tumor size was observed in the group treated with this inhibitor compared to control group (Figure 7D).

Overall, our findings demonstrate that miR-877-5p regulated TNBC cells’ viability of 4T1 cells. Moreover, our results show that miR-877-5p inhibition could possess therapeutic potential for the treatment of TNBC given that a single dose of a miR-877-5p was able to diminish tumor growth in 4T1-derived tumors. Also, miR-877-5p/TIMP3/IGF2 expression in BC tissue and/or plasma of patients could be a relevant molecular axis for BC development, with high relevance in patients with AAMS.

## 3. Discussion

In this study we provided new possible participants in the interaction between BC and AAMS based on miRNA detection. Dysregulation in miRNA expression has been linked to several pathologies, including cancer and metabolic disorders [20,21]. In fact, some studies have reported that MS induces aberrant expression of miRNAs in the liver, adipose tissue and blood [27,28,29]. We hypothesize that circulating miRNAs of women with AAMS could impact BC development and progression and, hence, could represent important small molecules for the prognosis and treatment of this disease. In this work, we address the first step in highlighting this issue by identifying altered miRNAs in the plasma of women with AAMS, which also have functions in BC. Thus, we demonstrated that some of the circulating miRNAs in patients with AAMS have functions related to cancer (AAMS-downregulated miR-23a-3p, -19b-3p, -181a-5p, -122-5p, -425-5p, -28-3p, -146a-5p, let-7b-5p and AAMS-upregulated miR-101-5p and -877-5p) and some of them are also dysregulated in BC tissue (Figure 8). Interestingly, we found that let-7b-5p and miR-28-3p are decreased in plasma from patients with AAMS and are also decreased in breast tumors, while miR-877-5p is increased in both pathologies (Figure 8).

The Let-7 miRNA family is known to be involved in carcinogenesis and tumor progression, among other processes. Let-7b is known as a tumor suppressor, which plays an important role in the activation of mammary stromal fibroblasts and their connection with tumor cells. Let-7b levels have been found to be lower in tumor-associated fibroblasts (CAFs) than fibroblasts from ANT [30]. In addition, in samples of breast tissue, a greater expression of this miRNA is correlated with a better clinical prognosis in patients with ER+ BC [31]. Low levels of let-7b-5p indicate a low rate of disease-free survival and total survival in patients with basal-like BC, according to an integrative study in preneoplastic breast tissue [32]. Concerning miR-28-3p, there are no studies investigating its functions in BC, but a tumor suppressor role was suggested in esophageal squamous cell carcinoma, since it is downregulated in tumor tissue [33]. MiR-877-5p has been proposed as a tumor suppressor in hepatocellular and gastric cancer [34,35,36,37]. The role of miR-877-5p in BC is a hardly unexplored field. In the context of BC, a recent study suggested that miR-877-5p could have an inhibitory role in the epithelial to mesenchymal transition of MCF-7 cells by inhibiting the expression of FGB, which induces EMT [38]. Thus, this report suggests that miR-877-5p could be involved in BC progression by inhibiting it at least in a luminal A BC model. The role of miR-877-5p in TNBC was not previously investigated. In our study, we found that miR-877-5p is associated with lower survival in BC patients, and that its expression is inversely correlated with overall survival in BC patients.

We also found that miR-877-5p is increased in basal-like breast tumors compared to the other molecular subtypes. This is interesting since a retrospective study found that MS was more prevalent in TNBC compared with the other BC subtypes [15]. Also, we showed that miR-877-5p expression is inversely correlated with overall survival in BC patients, so we find it to be an interesting miRNA for BC associated with metabolic disorders. Hence, we focus our studies on this miRNA. Moreover, using public databases, we demonstrated that miR-877-5p is also increased in plasma from BC patients compared to plasma from healthy donors (Figure 8).

To provide a possible molecular mechanism for miR-877-5p’s role in BC, we identified IGF2 and TIMP3 as validated target genes of miR-877-5p whose expression is decreased in BC tissue and is negatively correlated with the levels of this miRNA in the tumors (Figure 8). According to the bioinformatic tool used in this work to predict validated target genes, TIMP3, IGF2, ERBB2 and COL6A2 were described as miR-877-5p direct targets and were negatively regulated by high-throughput sequencing of RNA isolated by crosslinking immunoprecipitation (HITS-CLIP), a methodology that generates unbiased genome-wide maps of miRNA–mRNA interactions [39,40]. Despite these findings, miRNA regulation is a complex and intricate process that does not always occur by direct binding. Transcriptional and post-transcriptional regulation, as well as epigenetic modifications, are potential mechanisms of miRNA regulation. Understanding this indirect regulation of miRNAs is crucial for unravelling the complex regulatory networks that govern gene expression. Hence, further investigation into these mechanisms is needed to elucidate miR-877-5p interaction with these genes in TNBC.

The tissue inhibitor of metallopeptidase 3 (TIMP3) binds directly to the extracellular matrix via the proteoglycan heparan sulfate [41]. Since it controls the activity of metalloproteases, it is considered a suppressor of angiogenesis, invasion, and metastasis, as well as an inhibitor of apoptosis [42]. It has been reported that a decrease in TIMP3 expression, among other genes, associated with an increase in CD44 may constitute an indicator of the invasiveness of BC ER+ or ER− [43]. In addition, TIMP3 overexpression in hormone receptor-positive BC cells reduces tumor growth and invasion while inducing apoptosis in ER+ tumor cells [42]. One study revealed that TIMP3, which is frequently hypermethylated in infiltrating ductal carcinomas, is associated with increased tumor grade and nodal metastasis, along with increased expression of ER, progesterone receptor (PR), and Her2 [44]. Lastly, it has been observed that TIMP-3 is expressed by fibroblasts in BC, and this expression is associated with a higher rate of distant metastases [45]. On the other hand, a large amount of evidence suggests that the insulin/insulin-like growth factor (IGF) pathway is highly involved in BC [46]. However, there are few studies that have evaluated the role of this growth factor in TNBC, being mostly described in ER+ tumors, where it promotes tumor growth and progression [47]. It has been observed that the CAFs present in BC metastatic sites exert a pro-tumorigenic role, in part exerted by the high production of IGF2 [48]. Particularly, regarding the study of this molecule in TNBC, there are controversial results. It has been described that in vitro stimulation of MDA-MB-231 cells with IGF2 increased their migratory capacity [49]. In addition, it has been shown in vivo, in TN mammary tumor specimens, that IGF2, together with ERβ1, is significantly expressed in TNBC [50]. However, recently, through an in silico analysis of gene enrichment with mutations in metastatic TNBC tumors, using the TCGA and METABRIC databases, IGF2 was described as one of the deleted genes [51].

Our studies using clinical datasets suggested that miR-877-5p could have effects on BC development and/or progression. Considering that the only current therapeutic alternatives to TNBC are still surgery, chemotherapy and radiotherapy, while other molecular subtypes have other therapeutics options, we focused our studies in this molecular subtype for functional analyses [2,3]. Thus, we demonstrated that miR-877-5p inhibition caused a decrease in the viability of TNBC 4T1 cells. Even more, a single dose with nanoparticles containing a miR-877-5p inhibitor reduced tumor growth of 4T1 derived tumors. Altogether, our results suggest that miR-877-5p could impact TNBC development; thus miR-877-5p inhibition could be an interesting therapeutic approach. This study is the first report showing that miR-877-5p could be a possible link between TNBC and AASM and demonstrating that miR-877-5p has an effect on TNBC growth. Further studies are needed to explore the role of miR-877-5p in TNBC metastasis and its effect on TNBC associated with AAMS.

## 4. Materials and Methods

### 4.1. Patient Recruitment and Sample Processing

For volunteer recruitment, an ethics protocol was established with Hospital Militar Central (CABA, Argentina). Healthy women were asked to sign an informed consent (IC) to enroll them in the protocol and complete a survey on risk factors associated with BC. Women of any age with no history of cancer were eligible. A small sample of peripheral blood was taken and placed in a sterile RNAse-free tube with EDTA as an anticoagulant. In addition, other patient data were requested, such as age, weight, height, waist diameter, blood pressure, biochemical data (blood glucose, cholesterol, triglycerides), and medication history, which were completed on separate forms by the clinician. The plasma from participants was obtained using centrifugation of blood collected with EDTA, and retained in a −80 °C freezer exclusively for laboratory use and locked.

Volunteers were divided into two groups, the control group and the AAMS group, when presented with two or more of these conditions: body mass index (BMI) ≥ 25.00 kg/m^2^, waist diameter ≥ 82 cm or high blood pressure (systolic ≥ 120, diastolic ≥ 80).

### 4.2. RNA Isolation

Total RNA isolation was performed using Tri Reagent (Molecular Research Center, Cincinnati, OH, USA) as previously described [52,53]. Before extraction, 40 fmol of cel-39 synthetic miRNA was spiked to 400 µL of plasma.

### 4.3. miRNA Microarrays

Four samples (2 control and 2 AAMS) were generated by pooling miRNAs from 9 donors. They were hybridized to GeneChip^®^ miRNA 4.0 Array (Affymetrix, Santa Clara, CA, USA). Data normalization and analysis were performed using Expression Console™ Software 1.3.1 and Affymetrix^®^ Transcriptome Analysis Console (TAC) Software. Significance was determined using Rank Products analysis. A false significant gene (FSG) < 0.05%, Log fold change (Log FC) > 1.5 and *p* value < 0.01 were used to identify differentially expressed miRNAs.

### 4.4. Functional Enrichment Analyses

Functional enrichment analysis was performed as previously described [54]. To investigate the molecular functions of differentially expressed miRNAs in plasma of patients with AAMS factors, we used DIANA miRPath v3 tool (https://dianalab.e-ce.uth.gr/html/mirpathv3/index.php?r=mirpath Accessed on 12 September 2020) [55] for obtaining a list of experimentally validated target genes derived from DIANA-TarBase v7. For functional enrichment analysis, we employed KEGG pathways and Gene Ontology (GO) resources. The results were merged, selecting Pathways Union in order to obtain both a heatmap and a histogram. The top 20 of the statistically significant terms (*p*-value < 0.01) were selected. The heatmaps returned using DIANA miRPath v3 were used to analyze the relevance of each miRNA in each pathway.

### 4.5. TCGA Dataset Analysis

miRNA and gene expression analyses from TCGA were performed similar to those that have been described previously [56]. Briefly, clinical pathological data (PAM50 molecular subtype), mature miRNA and gene expression of BC and ANT of patients were obtained from TCGA breast cancer (TCGA-BRCA) cohort and from normal mammary gland (NMG) tissues from the GTEx project available in the UCSC Xena bioinformatics tool (https://xena.ucsc.edu/) [57]. The miRNA-Seq (IlluminaHiSeq_miRNASeq) and RNA-Seq (IlluminaHiSeq) data were downloaded as log2 (RPM + 1) values. A total of 75 BC samples were paired with 75 ANT samples, excluding patients without paired samples. To determine miRNA and gene expression levels based on PAM50 molecular subtypes and to analyze survival curves, we utilized all available tumor samples present in the TCGA-BRCA. The Shapiro–Wilk test, the Levene F test, and the box plot were used to assess data normality and variance homogeneity. Paired Sample t-test was applied for data that fulfilled the requirements using Sigma Plot 12.0. Otherwise, the Wilcoxon signed-rank test was performed. When three or more groups were analyzed, one-way ANOVA followed by the Tukey test was performed for data that met the requirements. Otherwise, the Kruskal–Wallis rank followed by Dunn’s test was performed. Kaplan–Meyer curve analysis was performed using UCSC Xena tool and the log-rank test was carried out to determine significance.

### 4.6. Correlation Matrix

Validated target genes of miR-877-5p that are involved in processes related to cancer were determined using KEGG with DIANA miRPath v3, and their expression levels were downloaded from TCGA BRCA data available in UCSC Xena, as described above. Using the Hmisc R package, we generated a correlation matrix for these miRNAs and their target genes, applying the Spearman correlation coefficient. Genes with a negative correlation with miR-877-5p expression (*p*-value < 0.05) and correlation coefficient rho <−0.2 were selected for further analysis.

### 4.7. miRNA RT-qPCR

miRNAs were retrotranscribed using the stem loop method as previously described [52,53] with some modifications. Briefly, 4 μL of total RNA in the case of plasma samples, was preheated in 14 μL containing 0.07 μM of stem loop primer at 70 °C for 5 min. Then, retrotranscription was performed in a final volume of 20 μL using M-MLV Reverse Transcriptase (Promega, Madison, WI, USA) and incubated in a TC 9639 Thermal Cycler (Benchmark, Sayreville, NJ, USA) for 30 min at 16 °C, 60 min at 42 °C and 2 min at 70 °C. qPCR was performed with FastStart Universal SYBR Green Master (Roche, Basel, Switzerland) using a final volume of 10 μL, 0.1 μM primers and 1 μL of cDNA. Reactions were incubated at 50 °C for 2 min, 95 °C for 10 min, 40 cycles of 95° C for 15 s, annealing temperature for 15 s, and 60 °C for 1 min. All reactions were run in duplicate. The expression levels of miRNAs were normalized to cel-39 levels. Primer sequences for miRNA RT-qPCR are as follows: RT-miR-877-5p Fw STEM, GTCTCCTCTGGTGCAGGGTCCGAGGTATTCGCACCAGAGGAGACCCCTGC; RT-cel-miR-39-3p Fw STEM, GTCTCCTCTGGTGCAGGGTCCGAGGTATTCGCACCAGAGGAGACCAAGCT; RT-cel-miR-39-3p Fw, CGGGGTCACCGGGTGTAAATC; RT-miR-877-5p Fw, GGGCGGGTAGAGGAGATGGC and RT-Stemloop Rv, TGGTGCAGGGTCCGAGGTATT.

### 4.8. Cell Culture and Transfections

4T1 cells were grown in RPMI medium (GIBCO) supplemented with 10% of fetal bovine serum (FBS) and antibiotics.

For miR-877-5p inhibition, 4T1 cells were plated in 60 mm plates with 80% confluence and transient transfected using 50nM of miR-877-5p inhibitor or NC5 negative control (IDT Technologies, Coralville, IN, USA) by polyethylenimine methodology (PEI 25kDa linear—PolySciences Inc., Warrington, PA, USA) with PEI to RNA ratio 5:1.

### 4.9. Cell Viability Assay

4T1 cells were transfected as described above. After 24 h, cells were harvested and 4000 cells per well were plated in 96-well culture plates and grown in RPMI supplemented with 1% FBS and antibiotics for 48 h. Cell viability was assayed using MTS (Cell-Titer-96-wells Aqueous non-Radioactive Cell-Proliferation Assay, Promega, Madison, WI, USA) according to manufacturer instructions [58].

### 4.10. TNBC Syngeneic Transplant Model

Eight-week-old female Balb/c mice (*n* = 10) were housed under pathogen-free conditions following IBYME’s animal care guidelines. Mice were inoculated subcutaneously (s.c.) into the mammary fat pad with 10^4^ 4T1 cells. The animals were checked 3 times a week, and tumor size was determined using a digital caliper. On day 13, when tumors were around 80 mm^3^, mice were randomly divided into 2 experimental groups and inoculated intraperitoneally (i.p.) with 0.73 nmol of miR-877-5p inhibitor or NC5 negative control (IDT Biotechnologies) complexed with PEI in a 5:1 ratio (PEI to oligo) similar to previous methods [59,60]. Briefly, 0.73 nmol of oligonucleotides were mixed with PEI (5:1) in a final volume of 250 μL per animal and incubated for 20 min before injection [52].

### 4.11. Statistical Analysis

Unless indicated for each experiment section, statistical analysis was performed using Sigma Plot 12.0 based on “*n*” values corresponding to independent experiments. Data normalization and homogeneity of variances were assessed using Shapiro–Wilk test and Levene F test or box plot, respectively. A *t*-test was applied for data that fulfilled the requirements using Sigma Plot 12.0. Otherwise, a non-parametric signed-rank test, Mann–Whitney, was performed to test significance for experiments with 2 experimental groups. When one factor and three or more groups were analyzed, one-way ANOVA followed by Tukey post hoc test was performed when the samples in the groups followed a normal distribution and homogeneity of variance. Otherwise, the Kruskal–Wallis rank followed by the Student–Newman–Keuls test was performed.

## Figures and Tables

**Figure 1 ijms-24-16758-f001:**
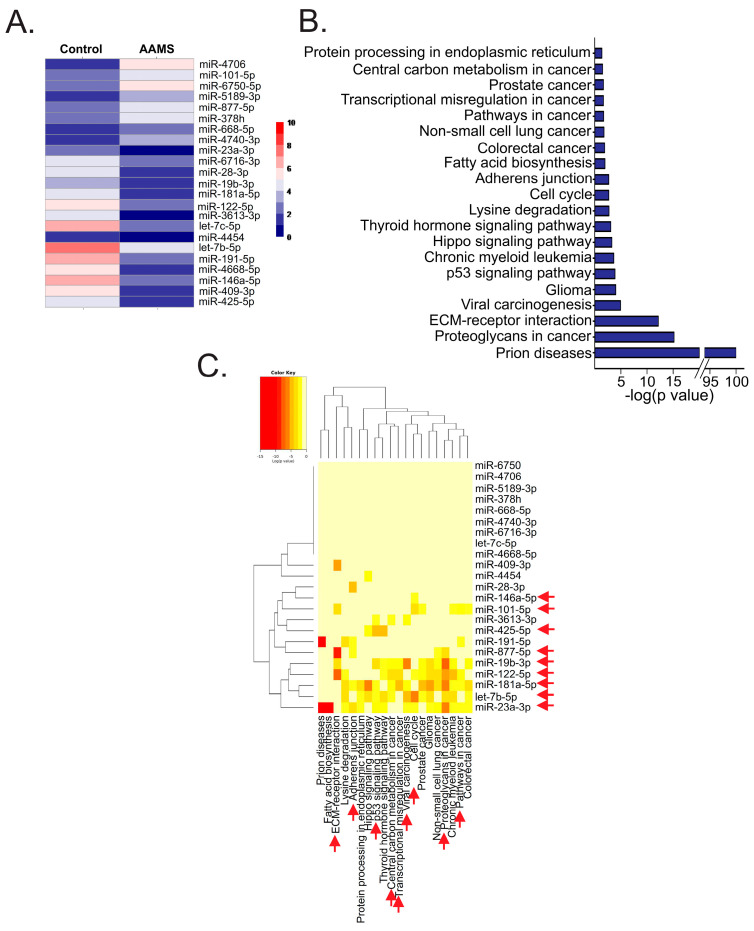
The expression of circulating miRNAs with functions related to cancer was altered in the plasma of women with AAMS. (**A**) MiRNAs were isolated from plasma of women with AAMS or the control group and hybridized to GeneChip^®^miRNA 4.0 Array (Affymetrix) (*n* = 2, each sample was generated by pooling the plasma from nine women). Heatmap depicting the differentially expressed miRNAs (FSG < 0.05%, LogFC > 1. 5 and *p* value < 0.01) is shown. (**B**) Histogram depicting the KEGG pathways regulated by the validated target genes from these miRNAs determined through DIANA miRPath. The top 20 most significant pathways are shown. (**C**) Heatmap + showing the miRNAs and their KEGG pathways. Red arrows indicate KEGG pathways relevant to cancer and the miRNAs with stronger association to these pathways.

**Figure 2 ijms-24-16758-f002:**
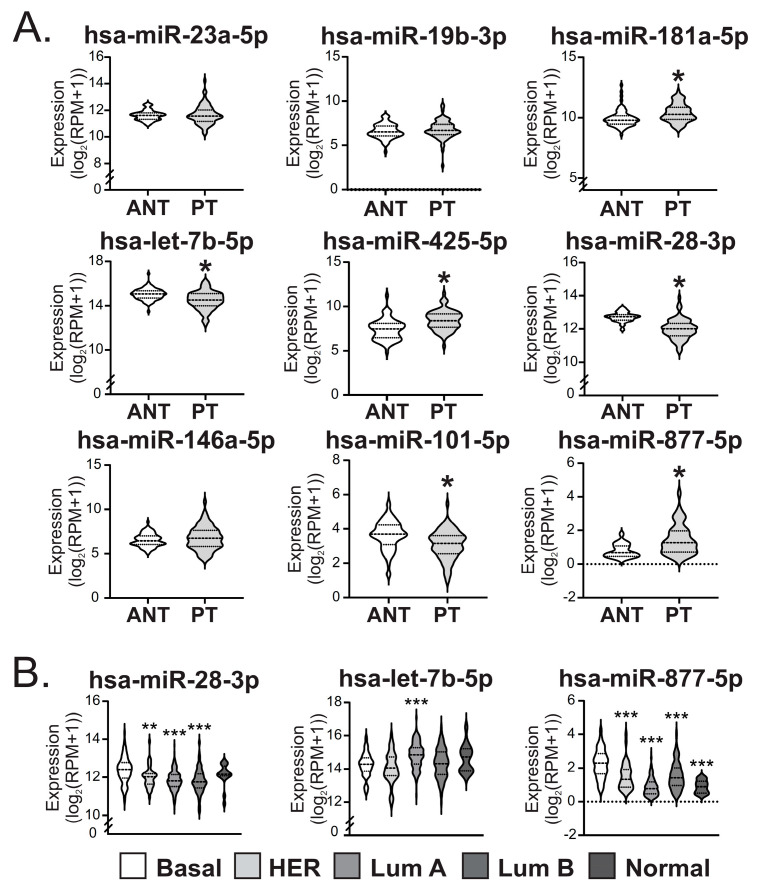
MiR-877-5p, -28-3p and let-7b-5p expression levels were altered in primary breast tumors from patients and their expression is dependent on the PAM50 subtype. (**A**) Expression levels of AAMS-modulated miRNAs were determined in primary tumors (PT) and adjacent normal tissue (ANT) of patients from TCGA BRCA data—Illumina HiSeq (*n* = 75). Significance was determined with paired t-test or Wilcoxon signed-rank test, when normality failed (*, *p* < 0.05). (**B**) Expression levels of miR-877-5p, miR-28-3p and let-7b-5p were determined in BC tissue from the different PAM50 molecular subtypes using TCGA BRCA data—Illumina HiSeq available in UCSC Xena tool (*n* = 1101). Basal means PAM50 basal-like BC; HER, HER2+ BC; Lum A, luminal A BC; Lum B, luminal B BC and Normal, normal-like BC. Significance was analyzed using one-way ANOVA or Kruskal–Wallis rank sum test, when normality failed. Asterisks indicate a significative difference compared to basal-like tumors (**, *p* < 0.01; ***, *p* < 0.001).

**Figure 3 ijms-24-16758-f003:**
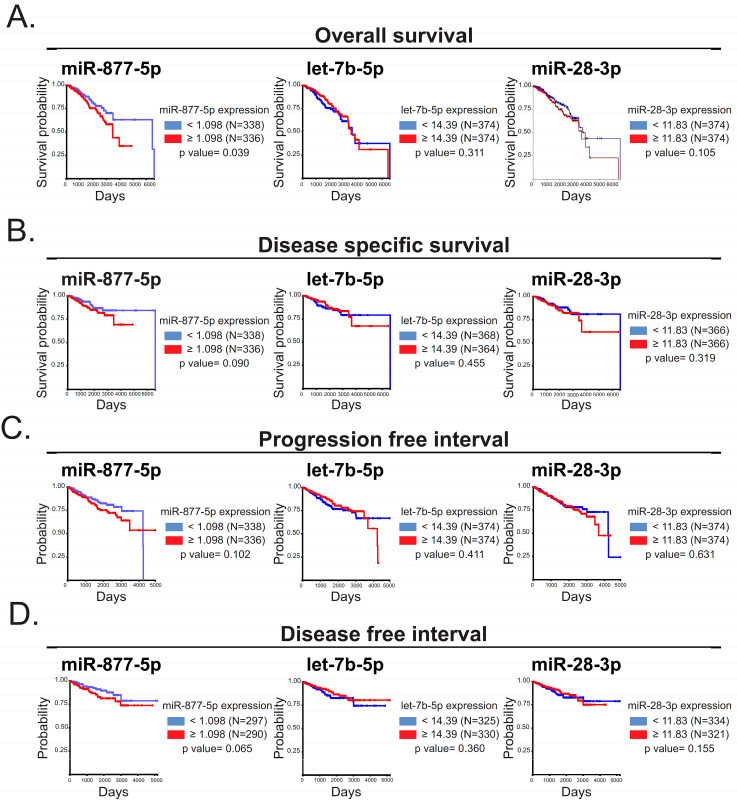
miR-877-5p expression was correlated with a diminished overall survival of patients. The impact of let-7b-5p, miR-28-3p and miR-877-5p in: (**A**) overall survival, (**B**) disease-specific survival, (**C**) progression-free interval and (**D**) disease-free interval of patients with BC was assessed in TCGA BRCA data—Illumina HiSeq using UCSC Xena tool. Log-rank test was carried out to determine significance.

**Figure 4 ijms-24-16758-f004:**
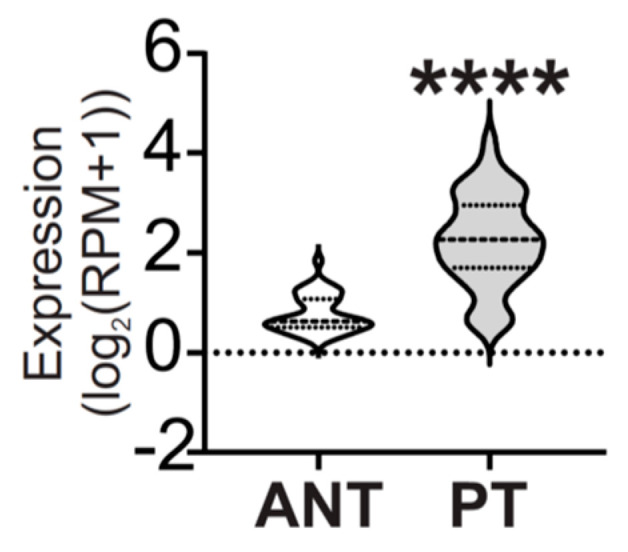
MiR877-5p expression levels were increased in primary tumors from patients with PAM50 basal-like BC. Expression levels of miR-877-5p were determined in primary tumors (PT) and adjacent normal tissue (ANT) of patients with PAM50 basal-like BC from TCGA BRCA data—Illumina HiSeq. Significance was determined using Mann–Whitney test (balanced N = 41; **** *p* < 0.0001).

**Figure 5 ijms-24-16758-f005:**
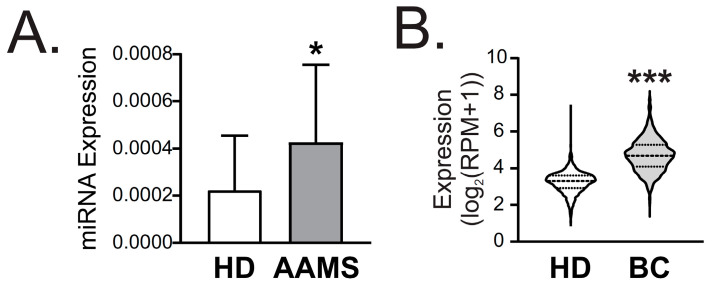
miR-877-5p expression was increased in plasma from patients with AAMS and from patients with BC. (**A**) miR-877-5 expression in plasma from women with AAMS or healthy donors (HD) (N = 18) was determined using stem loop RT-qPCR. Data were normalized to spike in cel-miR-39 and control group. Significance was evaluated using Mann–Whitney Wilcoxon signed-rank test (*, *p* < 0.05). (**B**) miR-877-5p expression in women with BC and HD was analyzed in the GSE73002 dataset (balanced N = 1280). Significance was evaluated using Mann–Whitney–Wilcoxon signed-rank test (***, *p* < 0.001).

**Figure 6 ijms-24-16758-f006:**
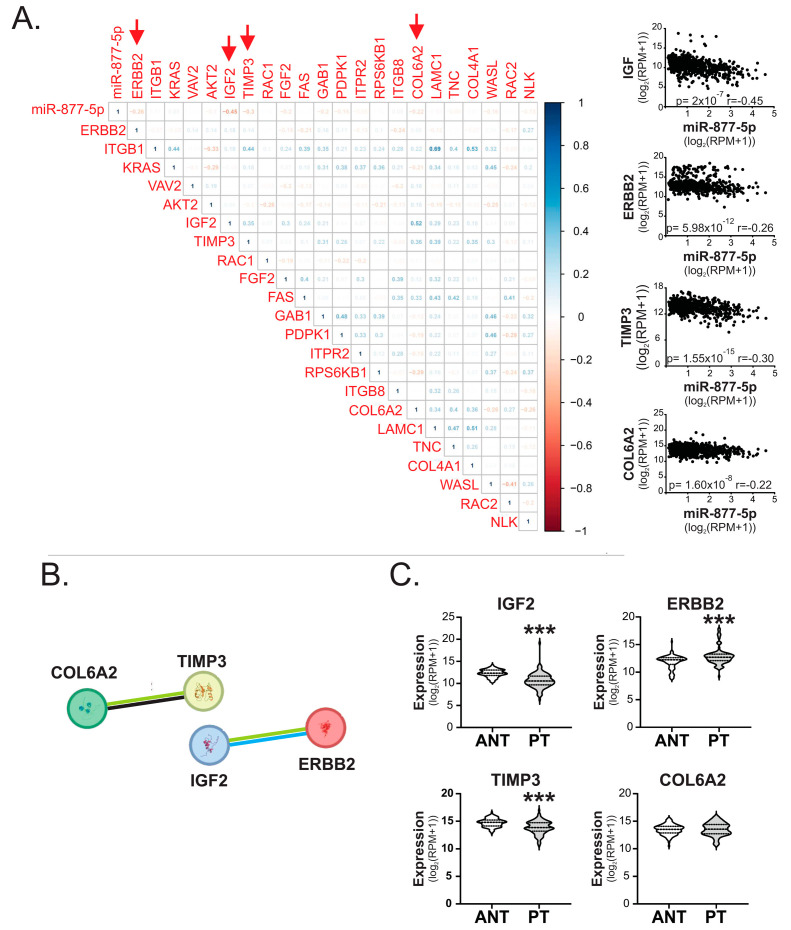
miR-877-5p expression levels were correlated with IGF2, ERBB2, TIMP3 and COL6A2 levels in mammary tumors. (**A**) Spearman test was performed to analyze correlation between miR-877-5p and its validated target genes in primary tumors of BC cohort TCGA BRCA data—Illumina HiSeq (*n* = 1101). Matrix correlation and dot plots of genes with low and moderate correlation are shown. Red arrows in the matrix indicate genes with low to moderate correlation with miR-877-5p. (**B**) Interaction network from STRING bioinformatic tool of the 4 genes with low to moderate correlation. Edges in black indicate co-expression; in green, text mining; and in blue, protein homology. The minimum required interaction score was 0.400 (medium confidence). (**C**) Expression levels of these genes in primary tumors (PT) and adjacent normal tissue (ANT) of patients from TCGA BRCA data—Illumina HiSeq (*n* = 75). Significance was determined using paired t-test or Wilcoxon signed-rank test, when normality failed (***, *p* < 0.001).

**Figure 7 ijms-24-16758-f007:**
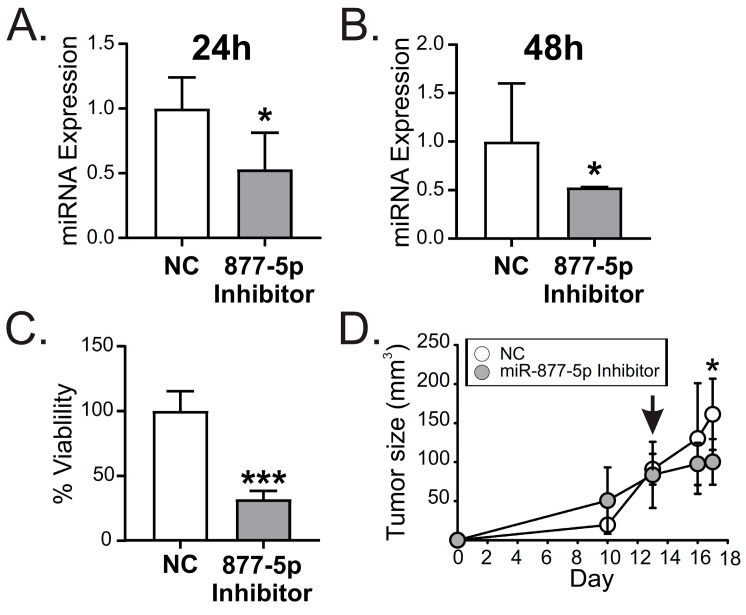
miR-877-5p inhibition decreased viability and adhesion in a model of TNBC. 4T1 cells were transfected with miR-877-5p inhibitor (877-5p inhibitor) or NC5 negative control (NC) (*n* = 3). (**A**) miR-877-5p expression was determined with RT-qPCR 24 h post-transfection or (**B**) 48 hours post-transfection. Data were normalized to U6 expression and control cells. (**C**) Twenty four hours post-transfection, cells were incubated with medium containing 1% FBS and cell viability was determined with MTS assay 48 h later. (**D**) Balb/c mice (*n* = 10) were inoculated with 4T1 cells s.c. On day 13, when the tumors were around 80 mm^3^, the animals were randomly divided and inoculated with PEI nanoparticles containing the miR-877-5p inhibitor or NC. Tumor size was determined with a digital caliper. The arrow indicates the day on which animals were treated with nanoparticles. Significance was determined using *t*-test (*, *p* < 0.05; ***, *p* < 0.001).

**Figure 8 ijms-24-16758-f008:**
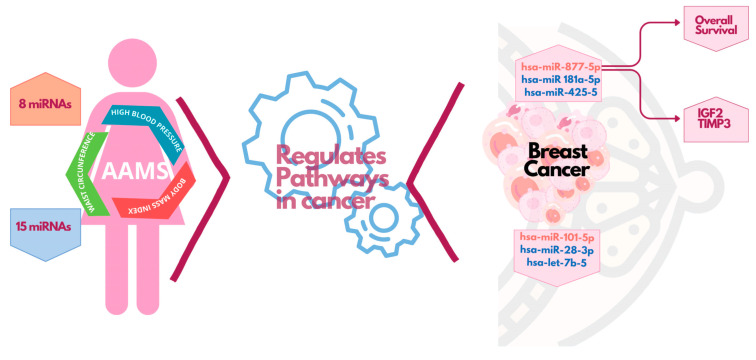
A cluster of circulating miRNAs dysregulated in plasma of women with AAMS has relevance in BC. Review diagram of the new insights presented in this paper. This diagram was performed using Canva free online resource (https://www.canva.com/ Accessed on 14 April 2023).

**Table 1 ijms-24-16758-t001:** Differentially expressed miRNAs in AAMS women compared to control (FSG < 0.05%, LogFC > 1. 5 and *p* value < 0.01).

Transcript_ID	miRNA	*p* Value	q Value	RP Values	LOG (FC)
MIMAT0019806	hsa-miR-4706	0	0.000	1.565	3.249
MIMAT0004513	hsa-miR-101-5p	2.17 × 10^−06^	0.005	6.774	2.136
MIMAT0027400	hsa-miR-6750-5p	2.17 × 10^−06^	0.005	4.738	2.348
MIMAT0027088	hsa-miR-5189-3p	1.09 × 10^−05^	0.013	10.029	2.316
MIMAT0004949	hsa-miR-877-5p	1.30 × 10^−05^	0.012	10.499	2.044
MIMAT0018984	hsa-miR-378h	3.69 × 10^−05^	0.028	13.581	1.720
MIMAT0026636	hsa-miR-668-5p	5.65 × 10^−05^	0.037	16.060	1.607
MIMAT0019870	hsa-miR-4740-3p	5.87 × 10 ^−05^	0.034	16.807	1.681
MIMAT0000078	hsa-miR-23a-3p	1.52 × 10^−04^	0.044	24.926	−2.008
MIMAT0025845	hsa-miR-6716-3p	1.04 × 10^−04^	0.032	21.203	−2.269
MIMAT0004502	hsa-miR-28-3p	9.56 × 10^−05^	0.031	20.584	−2.285
MIMAT0000074	hsa-miR-19b-3p	7.82 × 10^−05^	0.028	18.903	−2.342
MIMAT0000256	hsa-miR-181a-5p	5.87 × 10^−05^	0.023	16.699	−2.394
MIMAT0000421	hsa-miR-122-5p	6.52 × 10^−06^	0.003	7.366	−3.473
MIMAT0017991	hsa-miR-3613-3p	6.52 × 10^−06^	0.003	8.144	−3.599
MIMAT0000064	hsa-let-7c-5p	6.52 × 10^−06^	0.003	8.050	−3.415
MIMAT0018976	hsa-miR-4454	6.52 × 10^−06^	0.003	9.118	−3.258
MIMAT0000063	hsa-let-7b-5p	2.17 × 10^−06^	0.001	5.811	−3.629
MIMAT0000440	hsa-miR-191-5p	2.17 × 10^−06^	0.001	4.559	−3.631
MIMAT0019745	hsa-miR-4668-5p	2.17 × 10^−06^	0.001	5.091	−3.910
MIMAT0000449	hsa-miR-146a-5p	2.17 × 10^−06^	0.001	6.055	−3.531
MIMAT0001639	hsa-miR-409-3p	2.17 × 10^−06^	0.001	5.696	−3.495
MIMAT0003393	hsa-miR-425-5p	2-17 × 10^−06^	0.001	6.620	−3.550

**Table 2 ijms-24-16758-t002:** DIANA miRPath v3 validated target genes of miR-877-5p that are included in KEGG pathways related to cancer.

KEGG Pathway: Proteoglycans in Cancer (hsa05205)		
Gene Name	Gene Ensembl ID	Tarbase Methods
ERBB2	ENSG00000141736	HITS-CLIP
ITGB1	ENSG00000150093	HITS-CLIP
KRAS	ENSG00000133703	PAR-CLIP
VAV2	ENSG00000160293	HITS-CLIP
AKT2	ENSG00000105221	PAR-CLIP
IGF2	ENSG00000167244	HITS-CLIP
TIMP3	ENSG00000100234	HITS-CLIP
RAC1	ENSG00000136238	HITS-CLIP
FGF2	ENSG00000138685	PAR-CLIP
FAS	ENSG00000026103	HITS-CLIP
GAB1	ENSG00000109458	PAR-CLIP
PDPK1	ENSG00000140992	HITS-CLIP
ITPR2	ENSG00000123104	PAR-CLIP
RPS6KB1	ENSG00000108443	CLASH
**KEGG pathway: ECM-receptor interaction (hsa04512)**		
Gene Name	Gene Ensembl ID	Tarbase Methods
ITGB1	ENSG00000150093	HITS-CLIP
ITGB8	ENSG00000105855	HITS-CLIP
COL6A2	ENSG00000142173	HITS-CLIP
LAMC1	ENSG00000135862	Multiple
TNC	ENSG00000041982	HITS-CLIP
COL4A1	ENSG00000187498	Multiple
**KEGG pathway: Adherens junction**		
Gene Name	Gene Ensembl ID	Tarbase Methods
ERBB2	ENSG00000141736	HITS-CLIP
WASL	ENSG00000106299	HITS-CLIP
RAC2	ENSG00000128340	HITS-CLIP
NLK	ENSG00000087095	Multiple
RAC1	ENSG00000136238	HITS-CLIP

## Data Availability

The research data from microarrays have been submitted to GEO datasets (GSE235355).

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
