# Peer review of "miR-877-5p as a Potential Link between Triple-Negative Breast Cancer Development and Metabolic Syndrome"

_ijms, 2023, doi:10.3390/ijms242316758_

Round 1
Reviewer 1 Report
Comments and Suggestions for Authors
Moro et al. investigated the role of miR-877-5p in the progression of triple-negative breast cancer (TNBC) and its association with metabolic syndrome. They successfully identified IGF2 and TIMP3 as validated target genes of miR-877-5p and also discovered a miRNA inhibitor against miR-877-5p. Their findings indicate that inhibiting miR-877-5p could potentially serve as a therapeutic approach for treating TNBC.
The manuscript stands out for its clear and professional use of the English language, making it easily comprehensible. Moreover, the authors have skillfully organized their results section, using well-structured figures to present their data (although I have some additional comments on some of the figures). The authors have also created a comprehensive discussion section that delves into the molecular mechanism of miR-877-5p's role in breast cancer. Furthermore, the manuscript includes an up-to-date list of references. I have only some minor suggestions, which are outlined below:
(1) Please remove the extra spaces on lines 17, 145, and 257.
(2) Figure 6b appears to lack clarity and informativeness. It might be beneficial to either improve its resolution or redesign the figure. Alternatively, you could provide a more detailed description of the gene interactions within the main text or in the Figure 6 caption.
(3) Figure 3d is missing an axis label. Also, please keep the axis labels consistent in Figure 3.
(4) Just out of curiosity, in the experiment in Figure 3, why was such a long span of time selected (5000-6000 days)?
Author Response
Thank you for the clear review of our manuscript, your corrections helped us to greatly improve our manuscript.
A point by point response to the comments is given below:
(1) Please remove the extra spaces on lines 17, 145, and 257.
The extra spaces have been removed.
(2) Figure 6b appears to lack clarity and informativeness. It might be beneficial to either improve its resolution or redesign the figure. Alternatively, you could provide a more detailed description of the gene interactions within the main text or in the Figure 6 caption.
We agree with the reviewer, figure 6b is not clear and informative. To improve the presentation of this analysis we have modified the figure quality. In addition, we modified figure legend by adding:
“Edges in black indicate co-expression; in green, text-mining and in blue, protein homology. The minimum required interaction score was 0.400 (medium confidence).”
We have also changed the main manuscript by adding the following paragraph in line 199:
“Additionally, using the STRING database, we identified interactions among these genes (Figure 6B). This is reasonable considering all genes are involved in related pathways such as Proteoglycans in cancer (ERBB2, TIMP3 and IGF2) and ECM-receptor interaction (COL6A2). In particular STRING network revealed evidence of associations between COL6A2 and TIMP3 by text-mining and co-expression studies, while text-mining studies associate ERBB2 and IGF2 and also share protein homology (Figure 6B).”
Please notice that interactions provided by STRING for these 4 genes have changed, may be due to changes in automatic settings from this bioinformatic tool.
(3) Figure 3d is missing an axis label. Also, please keep the axis labels consistent in Figure 3.
We have added the axis label in figure 3d.
(4) Just out of curiosity, in the experiment in Figure 3, why was such a long span of time selected (5000-6000 days)?
The time selected for Kaplan-Meyer curves was provided by default for USCS Xena Browser tool. However the similar and significant results were obtained also when analysed survival correlation with miR-877-5p at 10 years.
Reviewer 2 Report
Comments and Suggestions for Authors
In the submitted research article, Moro et al. present their work to identify specific circulating miRNAs in patients with metabolic syndrome-associated alterations (AAMS) that may also impact breast cancer (BC). The researchers used microarray technology to detect miRNA alterations in the plasma of women with AAMS that are linked to cancer-related processes.
Metabolic syndrome (MS) elevates the risk of aggressive breast cancer, especially in the triple negative subtype (TNBC). This study explores microRNAs and their role in this connection. They found that miR-877-5p is increased in patients with MS and BC, linked to poorer survival. It's notably high in TNBC tumors. Inhibiting miR-877-5p reduced TNBC cell viability and adhesion, suggesting it could be a potential TNBC treatment.
After evaluating the manuscript, I have some comments and suggestions for the authors to consider:
Major comments:
- In figure 4, the authors state that miR-877-5p expression was diminished compared to NAT, while the data shows an increase in primary tumors. The authors should address this issue and clarify the correct interpretation of the obtained data.
- In figure 6, it would be useful to include a control for the correlations, showing a strong correlation coefficient. That would show that the presented data indeed shows low-moderate coefficient. It would also be necessary to include the used thresholds to determine when the correlation is considered high, moderate or low. In the same experiment, the figure shows that there is no difference in expression in COL6A2, this should also be mentioned and discussed in the text.
- The authors claim that miR-877-5p inhibition could have therapeutical potential in treating TNBC (lines 222-223). Are the viability and adhesion effect specific for TNBC cell lines? The authors should compare this effect with other cell lines (preferentially other BC and/or normal cell lines) to check if this inhibition affects them the same way.
Minor changes:
- Please use the metric system when defining the MS in the introduction (line 42).
- In line 78 the authors use waist diameter to determine AAMS. Is this correct, or is it the waist circumference?
- There are two consecutive "the" on line 90.
- The authors use the term "normal adjacent tissue" and "adjacent normal tissue" in an inconsistent way (e.g., they use NAT in the results section for figure 2, while they use ANT in the figure itself). Also, the definition of NAT is only present in the experimental section, and should also appear in the results the first time the term is used.
- The authors should define the different samples/conditions in figure 2B legend (e.g., HER, Lum A...), and standardize the way they show significance (either * or letters).
- In figure 5, HD should be defined in the figure legend.
- In general, figures could contain more labels in order to facilitate their understanding (e.g., titles for the categories in figure 3 and timing in figure 7).
- The authors should state the number of samples/experiments used in figures.
Author Response
Thank you for the clear review of our manuscript, your corrections helped us to greatly improve our manuscript.
A point by point response to the comments is given below:
-In figure 4, the authors state that miR-877-5p expression was diminished compared to NAT, while the data shows an increase in primary tumors. The authors should address this issue and clarify the correct interpretation of the obtained data.
I greatly appreciate the correction you have provided. Effectively miR-877-5p expression was augmented in basal-like tumors compared to ANT. We have corrected the main manuscript in line 155 by replacing "diminished” by “increased”.
- In figure 6, it would be useful to include a control for the correlations, showing a strong correlation coefficient. That would show that the presented data indeed shows low-moderate coefficient. It would also be necessary to include the used thresholds to determine when the correlation is considered high, moderate or low. In the same experiment, the figure shows that there is no difference in expression in COL6A2, this should also be mentioned and discussed in the text.
We appreciate your comment. To improve our manuscript, we have included the thresholds criteria that we used based on previous reports that were cited by modifying the sentence in line 194 to:
“The findings revealed a significant negative correlation between the expression levels of ERBB2, TIMP3, COL6A2, and IGF2 (Figure 6A). The correlation coefficient (rs) indicated a weak to moderate association between these genes. This classification of weak to moderate correlation coefficients (rs) was based on the criteria established by Guilford and Rowntree (weak: rs between -0.2 and -0.4; moderate: rs between -0.4 and -0.7)”.
Also the expression analyses of COL6A2 and ERBB2 was not described in Results so we replaced the paragraph in line 205 by:
“Moreover, we found that the expression of TIMP3 and IGF2 were diminished in the PT of BC patients from the BRCA-TCGA compared to NAT while ERBB2 expression was increased and COL6A2 is not differentially expressed in these tissues (Figure 6C).”
- The authors claim that miR-877-5p inhibition could have therapeutical potential in treating TNBC (lines 222-223). Are the viability and adhesion effect specific for TNBC cell lines? The authors should compare this effect with other cell lines (preferentially other BC and/or normal cell lines) to check if this inhibition affects them the same way.
We agree with the reviewer, the effects of miR-877-5p on cell viability and adhesion could be not specific for TNBC cell lines. Also, considering this miRNA is augmented in BC tumors compared to ANT (working with all molecular subtypes, Figure 1A) and our results in plasma of BC patients, we think that probably, miR-877-5p have effects in the other molecular subtypes. So, when we affirm that miR-877-5p could have therapeutic potential for TNBC, we think that this is relevant not because the effect is specific for TNBC, but because the gold standard treatment for TNBC relays still in chemotherapy and radiotherapy while the other molecular subtypes have other therapeutic options. That is why we focused on TNBC.
To clarify this issue, we have modified line 336 of the Discussion section by adding:
“Our studies using clinical datasets suggested that miR-877-5p could have effects in BC development and/or progression. Considering that the only current therapeutic alternatives of TNBC are still surgery, chemotherapy and radiotherapy, while other molecular subtypes have other therapeutics options, we focused our studies in this molecular subtype for functional analyses. Thus..”.
Minor changes:
- Please use the metric system when defining the MS in the introduction (line 42).
We have replaced 35 inches by 88.9 cm.
- In line 78 the authors use waist diameter to determine AAMS. Is this correct, or is it the waist circumference?
Thank you very much for the correction, we replaced diameter by circumference in MS definition.
- There are two consecutive "the" on line 90.
Thank you very much, we have deleted one “the” on line 90.
- The authors use the term "normal adjacent tissue" and "adjacent normal tissue" in an inconsistent way (e.g., they use NAT in the results section for figure 2, while they use ANT in the figure itself). Also, the definition of NAT is only present in the experimental section, and should also appear in the results the first time the term is used.
Thank you very much for these corrections, we have replaced normal adjacent tissue by adjacent normal tissue and its abbreviation NAT by ANT in lines 116, 117, 119, 121, 132, 156, 162, 206, 216, 390 and 395.
- The authors should define the different samples/conditions in figure 2B legend (e.g., HER, Lum A...), and standardize the way they show significance (either * or letters)
We have modified Figure 2B replacing the letters by asterisks to standardize the way to show significance. Also, legend from Figure 2B was modified to define the samples in figure 2B and the significance (**, p<0.01; **, p<0.001).
- In figure 5, HD should be defined in the figure legend.
The legend was corrected including that HD means healthy donors: healthy donors (HD).
- In general, figures could contain more labels in order to facilitate their understanding (e.g., titles for the categories in figure 3 and timing in figure 7).
Thank you for this suggestion. Figure 3 and 7 were modified to include categories of survival and the timing of Figure 7A and 7B.
- The authors should state the number of samples/experiments used in figures.
In line 104, we have added the number of samples from Figure 1A.
In line 132, we have added the number of samples from Figure 2A.
In line 135, we have added the number of samples from Figure 2B.
In line 212, we have added the number of samples from Figure 6A.
In line 217, we have added the number of samples from Figure 6B.
In line 229, we have added the number of samples from Figure 7.
Reviewer 3 Report
Comments and Suggestions for Authors
1. While elevated miRNA-877-5p has been found in MS as well as BC patients, these findings might be just correlative. The cytological evidence provided in this manuscript is not conclusive to provide causative evidence for miRNA-877-5p. Additionally, decrease in adhesion and viability might be inter-related.
2. It is also unclear whether decrease in adhesion might be detrimental or beneficial to cancer/metastatic potential while contrasting reports from Liu et al. (ref#35 ) suggesting miRNA-877-5p might have an inhibitory role in EMT warrants additional experiments. This could include migratory potential with miR-877-5p inhibition, testing in additional cell lines.
3. Please check for typos in the manuscript.
Comments on the Quality of English LanguageMinor typos and spelling errors detected. Nothing major and can be rectified.
Author Response
Thank you for the clear review of our manuscript, your corrections helped us to greatly improve our manuscript.
A point by point response to the comments is given below:
- While elevated miRNA-877-5p has been found in MS as well as BC patients, these findings might be just correlative. The cytological evidence provided in this manuscript is not conclusive to provide causative evidence for miRNA-877-5p. Additionally, decrease in adhesion and viability might be inter-related.
We agree with the reviewer that elevated levels of miR-877-5p in both, MS and BC, could be correlative and further studies must be done to elucidate if elevated levels of miR-877-5p in plasma could impact on breast cancer development and progression. However, we consider that this manuscript constitutes the first step to investigate the relation of miR-877-5p with MS and with BC. Considering the association we have presented, the next step is to demonstrate that miR-877-5p have a role in breast cancer cells phenotype. The role of miR-877-5p in BC was not previously explored and we demonstrate that miR-877-5p increases BC cell proliferation.
To improve our manuscript, we also include an in vivo experiment demonstrating that miR-877-5p inhibition decreases tumor growth of 4T1-derived syngeneic transplant.
Furthermore, we agree with the reviewer, the decrease in adhesion after miR-877-5p inhibition could be confusing to interpret without further experiments. We have performed cell migration studies, and we found that miR-877-5p inhibition has not modulated migration of 4T1 cells (supplementary figure is provided to reviewers).
We consider that further studies are needed. For example, in the future we will analyse if miR-877-5p regulates BC metastases in 4T1 syngeneic transplant model. We consider that we strongly demonstrate the role of miR-877-5p in BC growth but further studies are needed to elucidate the role of this miRNA in BC metastases.
To improve the manuscript, we decided to include our in vivo experiments and to remove cell adhesion and migration studies because an in vivo experiments are needed to elucidate the role in BC metastasis.
Also, we changed the title of the manuscript, considering we are not exploring miR-877-5p role in BC progression either in vitro or in vivo (we will investigate this in the future).
Finally, we enriched the discussion section to provide a clearer explanation that this study serves as the initial stage in determining whether the elevation of miR-877-5p in plasma due to MS could modulate BC development. The first step is to determine the expression levels in MS plasma and BC tissue and to determine if miR-877-5p modulate BC phenotype from TNBC models.
Line 2: miR-877-5p as a potential link between triple negative breast cancer development and metabolic syndrome. (we have replaced progression by development).
Line 30 to 33: The abstract was modified removing cell adhesion experiments and including in vivo results.
Line 73: We included the next paragraph “Finally, we demonstrated that miR-877-5p inhibition caused a decrease in viability of 4T1 TNBC cells. Even more, in vivo a single injection with a miR-877-5p inhibitor diminished tumor growth of 4T1-derived tumors.”.
Line 222: The result section title about 4T1 experiments was changed to “miR-877-5p inhibition decreased viability and tumor growth from the TNBC model 4T1”.
Line 223 to 249: This result section was modified to remove cell adhesion experiments and include in vivo experiments.
Line 227: Figure 7D was modified by replacing cell adhesion figure by tumor growth figure.
Line 233: Figure legend 7D was modified accordingly.
Line 346: The last paragraph was edited to explain that additional studies are necessary to determine the role of miR-877-5p in tumor progression as well as the effect of this miRNA on tumor development in a context of metabolic syndrome.
Line 444: Cell adhesion Section in Materials and methods was removed and a section about in vivo experiments was included.
- It is also unclear whether decrease in adhesion might be detrimental or beneficial to cancer/metastatic potential while contrasting reports from Liu et al. (ref#35 ) suggesting miRNA-877-5p might have an inhibitory role in EMT warrants additional experiments. This could include migratory potential with miR-877-5p inhibition, testing in additional cell lines.
Thank you very much for this suggestion. As we explained above, we agree with the reviewer, the decrease in adhesion after miR-877-5p inhibition could be confusing to interpret without further experiments. We have performed cell migration studies, and we found that miR-877-5p inhibition has not modulated migration of 4T1 cells (supplementary figure is provided to reviewers). We consider that further studies are needed. For example, in the future we will analyse if miR-877-5p regulates BC metastases in 4T1 syngeneic transplant model. We consider that we strongly demonstrate the role of miR-877-5p in BC growth but further studies are needed to elucidate the role of this miRNA in BC metastases.
To improve the manuscript, we removed cell adhesion results but included an in vivo experiment to focus our results in miR-877-5p in tumor growth.
Also, we improved the Discussion section.
Lines 285 to 287: The discussion section was edited to discuss the results previously shown by Liu et. al.
Line 340 to 341: The discussion regarding miR-877-5p effect on cell adhesion were removed.
Line 346: We have added the sentence “Further studies are needed to explore the role of miR-877-5p in TNBC metastasis and its effect in TNBC associated with AAMS.”.
3. Please check for typos in the manuscript.
Thank you very much, we have checked for typos and corrected them .

Reviewer 4 Report
Comments and Suggestions for Authors
In this study, the authors investigated the role of miR-877-5p in triple negative breast cancer progression and metabolic syndrome. The data showed that miR-877-5p was overexpressed in plasma from patients with AAMS and in BC tumors. And higher miR-877-5p expression was associated with lower patient survival. MiR-877-5p inhibition diminished viability and adhesion capability of TNBC 4T1 cells. Mechanically, miR-877-5p could target IGF2 and TIMP3, whose expression is decreased in BC tissue and moreover, is negatively correlated with the levels of this miRNA in the tumors. Overall, this study is interesting and my specific comments are listed below.
1. It stated: “MiRNAs were isolated from plasma and four samples (2 control and 2 AAMS) were generated by pooling miRNAs from 9 donors…”. Why did the authors pool the miRNAs from 9 donors? And how did the author perform statistical analysis if you only have 4 samples (2 vs 2)?
2. In Figure 2B, please describe the “abc”, “a”, “bd”, “ce”, “de”, “b”, “c”… stand for. And what’s the “p value < 0.05” for?
3. In both figure 2B and figure4, the miR877-5p expression are higher in basal-like breast patients. Why did the authors conclude that “MiR877-5p expression levels were decreased in primary tumors from patients with PAM50 basal-like BC”?
4. For figure 5, I don’t think violin plots can give us more information than the dot bar graphs. I recommend the authors should present the RT-qPCR data with bar graphs.
5. miR-877-5p expression was negatively correlated with IGF2, ERBB2, TIMP3 and COL6A2. Mechanically, the authors should demonstrate miR-877-5p can target IGF2 and TIMP3. The standard method could be predicting the miR-877-5p sites in IGF2 and TIMP3 genes and then perform luciferase assays to prove them.
6. For figure 7, I strongly recommend the authors should show us the representative images of viability and adhesion data, but not only the quantification results. Moreover, the violin plots are not encouraging for this kind of data presentation. Besides 4T1, one more cell line is required for viability and adhesion assays.
7. It’s better to have an animal model for validation of the function of miR-877-5p in triple negative breast cancer progression and metabolic syndrome.
Comments on the Quality of English LanguageI have no comments.
Author Response
Thank you for the clear review of our manuscript, your corrections helped us to greatly improve our manuscript.
A point by point response to the comments is given below:
- It stated: “MiRNAs were isolated from plasma and four samples (2control and 2 AAMS) were generated by pooling miRNAs from 9donors…”. Why did the authors pool the miRNAs from 9 donors? And how did the author perform statistical analysis if you only have 4 samples (2 vs 2)?
Thank you for your thoughtful question regarding our methodology. The decision to pool miRNAs from 9 donors into 4 samples (2 control and 2 AAMS) was made based on practical considerations and our study design. Pooling miRNAs from multiple donors into a single sample is a common approach in miRNA profiling studies, primarily to reduce biological variability and enhance the statistical power of our analysis. This allows us to obtain a more representative snapshot of the overall miRNA profile in the respective groups, which can be particularly beneficial when dealing with limited sample sizes.
In terms of statistical analysis, we acknowledge the limited number of samples in our study. To address this, we applied Rank Products analysis for its robustness in handling small sample size experiments. Rank Products is a non-parametric, rank-based method that doesn't rely on data distribution assumptions. It is particularly suitable for datasets with limited samples and suitable for miRNA microarray analysis. To further enhance the rigor of our analysis we used three Rank Product parameters: alpha p-value < 0.01, False Significant Genes (FSG) < 10%, and FSG < 0.05%. Particularly we focus on the miRNAs with False Significant Genes (FSG) < 0.05% and Log fold change (Log FC) > 1.5 depicted in Table 1. We have added a supplementary table to the manuscript that contains this information with a color-coding explanation (Supplementary table 1 was cited in line 89). We do acknowledge the limitations of our approach given the small sample size and pooling method. To address this, we pursued RT-qPCR validation experiments with a sample size of n = 18.
To provide further clarification, we have indeed integrated the necessary modifications into the method section of the manuscript (line 372 to 373).
- In Figure 2B, please describe the “abc”, “a”, “bd”, “ce”, “de”, “b”,“c”… stand for. And what’s the “p value < 0.05” for?
We agree with the reviewer, the letters to describe significance are confusing. To resolve this issue, we have modified Figure 2B replacing the letters by asterisks to standardize the way to show significance. Also, legend from Figure 2B was modified to define the significance by adding in line 138 de sentence: Asterisks indicate a significative difference compared to basal-like tumors indicate a p value ((**, p<0.01; **, p<0.001).
- In both figure 2B and figure4, the miR877-5p expression are higher in basal-like breast patients. Why did the authors conclude that “MiR877-5p expression levels were decreased in primary tumors from patients with PAM50 basal-like BC”?
Thank you very much for the correction. Effectively miR-877-5p expression was augmented in basal-like tumors compared to NAT. We have corrected the main manuscript in line 155 by replacing "diminished” by “increased”.
- For figure 5, I don’t think violin plots can give us more information than the dot bar graphs. I recommend the authors should present the RT-qPCR data with bar graphs.
Thank you very much for this suggestion. We have replaced violin plots from Figure 5A by bar graphs.
- miR-877-5p expression was negatively correlated with IGF2, ERBB2, TIMP3 and COL6A2. Mechanically, the authors should demonstrate miR-877-5p can target IGF2 and TIMP3. The standard method could be predicting the miR-877-5p sites in IGF2 and TIMP3 genes and then perform luciferase assays to prove them.
We agree with the reviewer, to proof that the genes are direct target of miR-877-5p y BC tissue, further experiments are needed, and a standard method is the luciferase assay.
In the original version of the manuscript, we found that TIMP3, IGF2, ERBB2 and COL6A2 are validated target genes of miR-877-5p using the bioinformatic tool DIANA-TarBase v7. According with this tool TIMP3, IGF2, ERBB2 and COL6A2 were described as miR-877-5p direct targets and negatively regulated by high-throughput sequencing of RNA isolated by crosslinking immunoprecipitation (HITS-CLIP) a methodology that generates unbiased genome-wide maps of miRNA-mRNA interactions (Balakrishnan I, Stem Cells, 2014 and Karginov, Gene Dev., 2013).
To further investigate the potential interaction between these genes, we used other computational tools to predict potential binding sites for miR-877-5p within the mRNA sequences. Due to the unavailability of reagents and time constraints, conducting a luciferase assay was infeasible.
Addressing IGF2 as a target of miR-877-5p, we found through miRTargeLink2.0 (Kern F, Nucleic Acids Research, 2021) that both IGF2 and IGFBP2 are targets. This was already reported by CLASH assay (A Helwak, Cell, 2013). Moreover, IGFBPs regulate insulin-like growth factor (IGF) transport and uptake, as described by the Reactome pathway in GeneTrail 3.0. IGF2BP2 is a predicted target of miR-877-5p with an exact match to positions 2-8 of the mature miRNA using the Target Scan tool (McGeary SE, Science, 2019).
ERBB2 has not been predicted as a direct target of miR-877-5p by other tools besides DIANA-TarBase v7. However, genes like KRAS have been predicted as an exact match to positions 2-8 of the mature miRNA using the Target Scan tool (McGeary SE, Science, 2019). It is described in the Reactome pathway GRB2 events in ERBB2 signalling. In addition, UBC and RNF41 are reported as miR-877-5p targets (A Helwak, Cell, 2013) and are involved in pathways that regulate ERBB2 signalling, specifically in the downregulation of ERBB2 signalling and downregulation of ERBB2:ERBB3 signalling.
Despite these findings, we are aware that miRNA regulation is a complex and intricate process that does not always occur by direct binding. Transcriptional and post-transcriptional regulation, as well as epigenetic modifications, are potential mechanisms of miRNA regulation. Understanding this indirect regulation of miRNAs is crucial for unravelling the complex regulatory networks that govern gene expression. However, it is important to note that further investigation into these mechanisms goes beyond the scope and aim of this particular project.
To improve our manuscript, we have added a paragraph in line 301 of Discussion section.
- For figure 7, I strongly recommend the authors should show us the representative images of viability and adhesion data, but not only the quantification results. Moreover, the violin plots are not encouraging for this kind of data presentation. Besides 4T1, one more cell line is required for viability and adhesion assays.
Thank you very much for this correction. The violin plots in Figure 7 were replaced by bar graphs . Viability assays were performed by MTS according to manufacturer instruction. In this type of methodology, mitochondrial activity is measured as an indirect determination of cell viability. The presence of oxidised formazan is detected with a lector of absorbance for 96-well plates, so it is not possible to have a representative picture.
In response to another reviewer's comments, we have made the decision to remove the adhesion data experiment from the manuscript. The interpretation of this data without additional experiments could potentially lead to confusion. We have performed cell migration studies, and we found that miR-877-5p inhibition has not modulated migration of 4T1 cells (supplementary figure is provided to reviewers). We consider that further studies are needed. For example, in the future we will analyse if miR-877-5p regulates BC metastases in 4T1 syngeneic transplant model.
To improve the manuscript, we decided to focus this paper on miR-877-5p role in tumor growth including in vivo experiments and to remove cell adhesion and migration studies because an in vivo experiments are needed to elucidate the role in BC metastasis.
Also, we changed the title of the manuscript, considering we are not exploring miR-877-5p role in BC progression either in vitro or in vivo (we will investigate this in the future).
Hence, we modified the manuscript as described below:
Line 2: miR-877-5p as a potential link between triple negative breast cancer development and metabolic syndrome. (we have replaced progression by development).
Line 30 to 33: The abstract was modified removing cell adhesion experiments and including in vivo results.
Line 73: We included the next paragraph “Finally, we demonstrated that miR-877-5p inhibition caused a decrease in viability of 4T1 TNBC cells. Even more, in vivo a single injection with a miR-877-5p inhibitor diminished tumor growth of 4T1-derived tumors.”.
Line 222: The result section title about 4T1 experiments was changed to “miR-877-5p inhibition decreased viability and tumor growth from the TNBC model 4T1”.
Line 223 to 249: This result section was modified to remove cell adhesion experiments and include in vivo experiments.
Line 227: Figure 7D was modified by replacing cell adhesion figure by tumor growth figure.
Line 233: Figure legend 7D was modified accordingly.
Line 346: The last paragraph was edited to explain that additional studies are necessary to determine the role of miR-877-5p in tumor progression as well as the effect of this miRNA on tumor development in a context of metabolic syndrome.
Line 444: Cell adhesion Section in Materials and methods was removed and a section about in vivo experiments was included.
- It’s better to have an animal model for validation of the function of miR-877-5p in triple negative breast cancer progression and metabolic syndrome.
Thank you very much for this revision. As was mentioned above, we have included an in vivo experiment where we showed that miR-877-5p inhibition caused a decrease in viability of 4T1 TNBC cells. Even more, in vivo a single injection with a miR-877-5p inhibitor diminished tumor growth of 4T1-derived tumors.
Also, we consider that this manuscript constitutes the first step to investigate the relation of miR-877-5p with MS and with BC. Considering the association we have presented, the next step is to demonstrate that miR-877-5p have a role in breast cancer cells phenotype. The role of miR-877-5p in BC was not previously explored and we demonstrate that miR-877-5p increases TNBC cell proliferation and tumor growth. In future studies we will investigate miR-877-5p role in TNBC in a context of MS.
Hence, in line 346 we added a sentence explaining that additional studies are necessary to determine the role of miR-877-5p in tumor progression as well as the effect of this miRNA on tumor development in a context of metabolic syndrome.

Round 2
Reviewer 2 Report
Comments and Suggestions for Authors
The authors have provided convincing explanations to the my questions and have added valuable data to the study. With the new incorporated data and changes, I believe it can be published in the present form.
Author Response
Thank you very much again for the clear review of our manuscript. We greatly appreciate all the corrections you have made to our manuscript.
Reviewer 3 Report
Comments and Suggestions for Authors
Thank you for the changes.
miR-877-5p inhibitor seems to have surprisingly controlled tumor in vivo really quickly and with just one dose.
In line 448, you suggest that animals were treated when tumors were palpable on day 13. However, your graph suggests the tumors were around 80mm^3 on day 13 when treatments were administered. Mammary fat pad tumors are palpable way before they reach these volumes. So, the timeline you mentioned for these experiments seems not accurate.
Author Response
Thank you very much for this correction. We agree with the reviewer, the treatment was performed when tumors have around 80mm3 and all mice have tumors palpable and measurable. Thus, we have corrected the manuscript as is detailed below in line 234, 240 and 448 by replacing the word “palpable” by “around 80 mm3”.
Reviewer 4 Report
Comments and Suggestions for Authors
The authors revised their manuscript and provided the relatively reasonable explanations to my questions. Some details were also added in this manuscript. The data in this study are integrated and more convincing. This manuscript should be able to bring the readers some new understanding about the role and mechanism of miR-877-5p as a potential link between triple negative breast cancer progression and metabolic syndrome. I have no any other questions for the current version.
Author Response

(The authors gave the same response as above.)

Round 3
Reviewer 3 Report
Comments and Suggestions for Authors
Thank you for the changes. The manuscript looks more coherent now. No further comments.